# Genetic Diversity and Population Structure of Moroccan Beni Ahsen: Is This Endangered Ovine Breed One of the Ancestors of Merino?

Asmae Kandoussi [1,2], Ismaïl Boujenane [1], Mohammed Piro [3] and Daniel Pierre Petit [2,*]

[1] Department of Animal Production and Biotechnology, Institut Agronomique et Vétérinaire Hassan II, P.O. Box 6202, Rabat 10101, Morocco; asmae.kandoussi@gmail.com (A.K.); ismail.boujenane@gmail.com (I.B.)

[2] Glycosylation et Différenciation Cellulaire, UR 22722, Laboratoire Peirene, Université de Limoges, 123 av. A. Thomas, CEDEX, 87060 Limoges, France

[3] Department of Medicine, Surgery and Reproduction, Institut Agronomique et Vétérinaire Hassan II, P.O. Box 6202, Rabat 10101, Morocco; vetpiro@yahoo.fr

* Correspondence: daniel.petit@unilim.fr

**Abstract:** (1) Background. Merino is a worldwide sheep breed well known for the quality and quantity of its wool. If there is no doubt that it originates from the Mediterranean Basin and that human selection took place in southern Spain, the populations that potentially contributed to the building of this breed are a matter of debate. Here, we tested whether a Moroccan breed settled on the North Atlantic coast, the Beni Ahsen, could be a good candidate, given the thickness and distribution of fleece covering the head and legs. (2). Methods. Using the control region of the mtDNA, 20 Beni Ahsen sequences were considered in a dataset of 643, including Mediterranean Merino and non-Merino breeds. Unfortunately, the Beni Ahsen is an endangered breed because of a lack of interest from the breeders. (3) Results. European Merino-derived breeds are divided into an Iberian and an Italian cluster, more linked to non-Merino breeds of the same country than between Merino themselves. Beni Ahsen breed is strongly linked to the other Moroccan breeds but shows the greatest number of connections with Merino-derived breeds, especially from Iberia. Interestingly, several other Moroccan breeds are also connected to Iberian Merino.

**Keywords:** Beni Ahsen; Merino; mtDNA; control region

## 1. Introduction

There is no doubt that the cradle of Merino sheep takes place in the south of the Iberian Peninsula [1,2]. On the basis of ovine votive offerings showing the wavy wool, Pedrosa et al. [3] argued that Merino dated at least c. 300–200 BCE in southern Spain. The Roman author Columella of the first century AD reported that fine wool animals from Tarento (the south of Italy), also named the Greek breed, were crossed with more coarse wool animals from North Africa to improve the wool characteristics of the local animals of the Cadix area near the Strait of Gibraltar [4]. Later, it is generally agreed that the Moors, who dominated Spain from the eighth to thirteenth centuries, were primarily responsible for selectively breeding the animals to such an extent that the wool they produced became superior to that of all other sheep. Given the historical documentation cited above, it is unlikely that they imported animals with Merino characteristics at that time, contrary to the opinion sometimes claimed [5]. However, the historical data give a complex origin of populations involved in the construction of early Merino animals, given the probable role played by Phenicians, Romans, and Moors [6]. If it is admitted the involvement of North African sheep in the genitors of Merino, the question of the most probable breeds is open. H. Geoffroy Saint-Hilaire [6] indicated that among the population recorded in 1929, several characters specific to the Merino phenotype (chest dewlap, dense fleece making waves, wool covering the forehead and cheeks) could be retrieved in Beni Guil (variety

Harcha according to Boujenane [7]) and in the animals of the great plains of west-center Morocco [8–10]. This last group corresponds to the range area occupied nowadays by Sardi, Boujaad, and Beni Ahsen breeds (Figure 1). After a century, selection has shaped these animals toward standardized breeds, and the Merino phenotype remains recognizable in Beni Ahsen settled in Gharb plain (Table S1) and to a lower extent in Beni Guil (oriental part of Morocco).

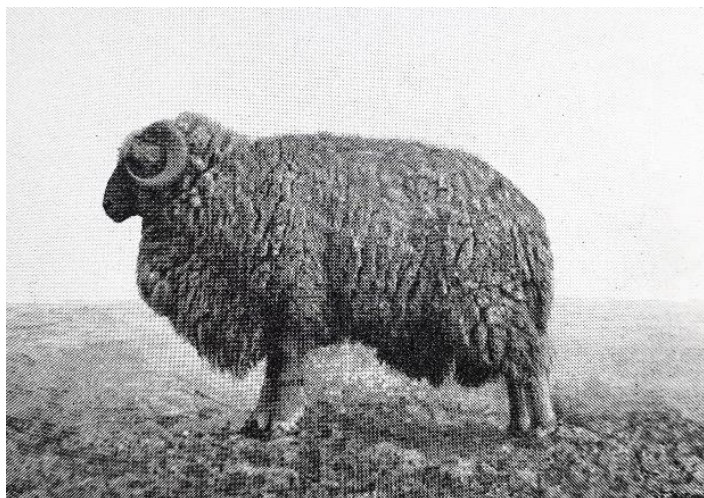

**Figure 1.** Beni Ahsen ram in 1929 (after [4]).

Unfortunately, Beni Ahsen is an endangered breed [7,11,12] for two reasons: (i) its fattening faculty is low, and (ii) during the dry years, breeders of neighboring areas used to introduce their animals (Zemmour, Timahdite) in the Gharb plain to take advantage of humidity and thus of more abundant feed, resulting in competition and crossing possibilities between breeds. The situation is even worse now because of global warming and the preference of breeders for more productive breeds than Beni Ahsen. It explains why the ANOC (Caprine and Ovine National Association) did not consider this breed in their improvement program conducted for the other breeds [7]. The same author reported that the population size at this date was about 385 thousand heads, i.e., around 2% of the total sheep population. In the last decade, the population size of pure animals continued dramatically to decline, reaching certainly less than 1000 individuals in 2019.

Due to the difficulty of finding purebred individuals, Beni Ahsen was not included in the previous works dedicated to the main breeds of Morocco using the sequence of the control region of mitochondrial DNA [13,14]. In the present work, the enrichment of our database with new sequencing data of Beni Ahsen individuals allowed us to address several points. The first purpose was to quantify the polymorphism of the Beni Ahsen breed in order to test whether there is a risk of genetic erosion due to its small population size. The second one was to investigate the relationship between Moroccan breeds through different methods. The third one was to calculate the degree of proximity between the major Moroccan breeds with Merino and Merino-derived animals. The crucial question was to test whether maternal markers support the idea emitted by Bourfia [11], according to which the Beni Ahsen would be the best Moroccan candidate as one of the ancestors of Merino.

## 2. Materials and Methods

A total of 20 blood samples were collected in 2019 from farmers in the breeding area of Beni Ahsen by an authorized veterinary person during a follow-up. This work was approved by the Ethical committee of Hassan II and Agronomy Veterinary Institute of Rabat on 24 February 2022. We took care that only one or two animals were chosen within a flock in order to avoid related specimens. Moreover, we only considered individuals corresponding to the standard of the breed [11,12] to reduce the impact of introgression

with foreign commercial or other local breeds. Breeders consented to these samplings and their utilization for scientific purposes. Blood samples were taken from the jugular vein and collected in ethylene diamine tetra acetic (EDTA) acid tubes and frozen at −20 °C until extraction. Total genomic DNA was extracted from blood samples using the alkaline lysis method. The purified extracts of amplified DNA were treated using an ABI Prism 3100 Genetic Analyzer (Applied Biosystems, Waltham, MA, USA). Details about PCR amplification and sequencing are reported in Kandoussi et al. [13]. The control region of mtDNA of every Beni Ahsen animal was sequenced on both strands to avoid errors, from nucleotides 15453 to 16045 of the reference sequence AF010406 [15], and then recorded in GenBank (accession numbers OM423782 to OM423801).

The data concerning the other Moroccan breeds (Sardi, Timahdite, Boujaad, Beni Guil, D'man, and Blanche de Montagne) were taken from GenBank (accession numbers MN229085 to MN229277).

Using the DNAsp v5 program [16], several diversity parameters were calculated to estimate the genetic variations of Moroccan breeds: the number of haplotypes (H), the haplotype diversity (Hd), the nucleotide diversity (π) were obtained with the "DNA Polymorphism" option. The number of segregation sites (S), singletons (Sg) and parsimoniously informative sites (P) were computed using the "Polymorphic sites" option. The significance of the values corresponding to each breed was established by using a Student t-test for unequal variances implemented in PAST program version 2.17c [17]. To test possible selection or population expansion, four neutrality tests were conducted: Fu and Li F*, Fu and Li D*, and Fu Fs and D of Tajima using DNAsp for each Moroccan breed.

To create the sequence database, in addition to Moroccan breeds, we took each of the Beni Ahsen sequences to explore the NCBI Gene Bank using the BLASTn algorithm and selected all the sequences sharing the smaller number of mutations. Seven breeds affine to Merino were retrieved (Spanish Merino, Merino Preto, German Merino, Transylvanian Merino, Sopravissana, Merinizzata, and Gentile di Puglia) [18]. We also found breeds qualified as "primitive" (Polish Swiniarka) or from territories of the Eastern Mediterranean (Awassi) and beyond (Tibetan sheep). The dataset was then enriched with breeds found using the other Moroccan breeds as blast seeds [13]: Campanica, Algarvia, Churra T. Quente, and Latxa Black Face from Iberia, Appenninica, Laticauda and Comisana from Italia. We finally added two primitive breeds, the Corsica (*Ovis musimon*) and the Anatolia Mouflons (*Ovis gmelini gmelini*). For each breed found, 25–30 sequences were recorded when available (Table 1).

**Table 1.** List of breeds considered in the database. The bold characters correspond to Merino-related breeds. The population sizes are indicated in brackets. The access numbers are gathered in Table S2.

| Morocco | Iberia | Italia | Oriental and Primitive | Other |
|---|---|---|---|---|
| D'man (27) | Churra de Terra Quente (25) | Appenninica (29) | Polish Swiniarka (25) | **German Merino (29)** |
| Bl. de Montagne (32) | Churra Algarvia (30) | Lacaune of Italy (30) | Tibetan sheep (35) | **Transylvanian Merino (6)** |
| Boujaad (31) | Latxa Black Face (31) | Comisana (30) | Awassi (3) | |
| Sardi (33) | Campaniça (20) | **Gentile de Puglia (30)** | Anatolia Mouflon (3) | |
| Timahdite (37) | **Merino Preto (10)** | **Sopravissana (30)** | Corsica Mouflon (2) | |
| Beni Guil (31) | **Spanish Merino (26)** | **Merinizzata (30)** | | |
| Beni Ahsen (20) | | | | |

To analyze the degree of differentiation between Moroccan studied breeds, the pairwise FST values and their significances (*p*-values) were calculated between all combinations. Computation was performed with the hierstat package [19], using pairwise.neifst (for FST values) and boot.ppfst (for *p*-values) functions, while the overall FST value was calculated using the basic.stats function implemented in the same package. Nei's genetic distances between the breeds were calculated using the dist.genpop function implemented in the

adegenet package [19]. These distance values were considered to construct neighbor-joining trees using the PAST program version 2.17 [17].

Discriminant analysis of principal components (DAPC), implemented in the adegenet package R [20], was applied to the mtDNA dataset to examine the genetic structure of the populations and to assess the degree to which breeds differ from each other when considering prior information on breeds. The DAPC approach is assumed to optimize the separation of individuals into predefined groups based on a discriminant function of principal components. Moreover, DAPC analysis was used to assign individuals and to obtain the membership probability, which presents the overall genetic background of an individual.

All the retained sequences were used to construct a phylogenetic tree using MEGA software [21]. The phylogenetic tree was constructed using the Minimum Evolution (ME) method. The rate variation among sites was modeled with a gamma distribution (shape parameter = 5). The ME tree was searched using the Close-Neighbor-Interchange (CNI) algorithm at a search level of 1. The analysis involved 643 nucleotide sequences. All positions with less than 85% site coverage were eliminated, resulting in a total of 594 positions in the final dataset. The topology obtained was exported in Newick format. A program written in R (Newick-Extra [13]) counted the numbers of terminal branches that gather all the combinations between all the breeds. Thus, we know exactly how many times the Beni Ahsen are sister sequences to Spanish Merino, to Merinizzata, and so on, and ditto for the other Moroccan breeds. The advantage of this program is the possibility of calculating affinities between sequences without the constraints of different haplogroups. There is a type 1 affinity, in which only sister sequences are counted, and a type 2, where a sequence is a sister to two or more sister sequences (see supplementary data in [13] for details). Relationships between the studied breeds (asymmetric matrix generated by Newick-Extra) were visualized through the Non-Metric Dimensional Scaling Plot (NMDS) and Cluster analysis using Pearson correlation as distance metrics, using PAST software v.2.17 [17].

The aligned sequences were also submitted to the median-joining network implemented in Network software 10.2.0.0 version [22]. The "Star Contraction" option was used to reduce the large dataset, and the "MP calculation" procedure according to the neighbor-joining method was used to remove unnecessary vectors and median links and to avoid reticulations.

## 3. Results

### 3.1. Relationships between Moroccan Breeds

3.1.1. Polymorphism of Beni Ahsen Compared to Other Moroccan Breeds

Relative to the sampling number, Beni Ahsen presents variability parameters comparable to the other Moroccan breeds (Table 2). As regards neutrality tests, Beni Ahsen is situated in the penultimate position, just before the D'man, indicative of weak selection pressure or population expansion in the past.

3.1.2. Affinity Degree of Beni Ahsen to Other Moroccan Breeds

As shown in Table 3, the less differentiated breeds from Beni Ahsen, given by Fst values, are Sardi and Timhadite, with 0.00198 and 0.00294, respectively. The same result with the same couple of breeds was obtained with the Nei's genetic distances with 0.0062 and 0.0070, respectively.

To have a more comprehensive view of these data, a cluster analysis using Nei's genetic distances and Chord as a measure of distance was performed. It confirms the affinity of Beni Ahsen with the Sardi and Timahdite breeds, but this group must be enriched with the Blanche de Montagne breed. In contrast, the Boujaad breed is the most distantly related to the rest of the Moroccan breeds (Figure 2a). The same topology was obtained with Spearman's rho and Morisita index.

**Table 2.** Variability parameters and neutrality tests in seven Moroccan breeds. N: sampling number; S: number of segregation sites; Sg: number of singletons; P: number of parsimoniously informative sites; H: number of haplotypes; Hd: haplotype diversity; $\pi$: nucleotide diversity. Statistical significance: *: $p < 0.05$; **: $p < 0.01$; ***: $p < 0.0001$.

| | N | S | P | Sg | H | Hd | Π | Fu-Li's F * | Fu-Li's D * | Fu's Fs | Tajima D |
|---|---|---|---|---|---|---|---|---|---|---|---|
| Beni Ahsen | 20 | 65 | 22 | 43 | 19 | 0.995 | 0.02011 | −2.367 NS | −2.148 NS | −9.096 *** | −1.75292 NS |
| Bl. de Mont. | 32 | 44 | 16 | 28 | 24 | 0.97 | 0.0133 | −3.016 * | −2.943 * | −13.24 *** | −1.766 NS |
| Boujaad | 31 | 91 | 30 | 61 | 29 | 0.994 | 0.0226 | −3.280 ** | −3.105 * | −18.224 *** | −2.111 * |
| Beni Guil | 31 | 66 | 24 | 42 | 29 | 0.996 | 0.0164 | −2.931 * | −2.741 * | −22.713 *** | −1.953 * |
| D'man | 27 | 41 | 14 | 27 | 21 | 0.963 | 0.0151 | −0.880 NS | −0.634 NS | −7.424 * | −1.744 NS |
| Sardi | 33 | 76 | 42 | 34 | 30 | 0.994 | 0.0203 | −1.821 NS | −1.457 NS | −18.582 *** | −1.692 NS |
| Timahdite | 37 | 56 | 27 | 29 | 32 | 0.988 | 0.0215 | −2.502 NS | −2.242 NS | −25.845 *** | −1.855 * |

**Table 3.** Fst (above the diagonal) and genetic distances (below the diagonal) between the Moroccan breeds. The Fst values are all significant at the risk of 1%.

| Genetic Dist\FST | D'man | Blanche de Montagne | Boujaad | Sardi | Timahdite | Beni Guil | Beni Ahsen |
|---|---|---|---|---|---|---|---|
| D'man | 0 | 0.0622 | 0.1119 | 0.052 | 0.0517 | 0.0114 | 0.0878 |
| Bl. de Montagne | 0.0082 | 0 | 0.0852 | 0.0233 | 0.0223 | 0.0193 | 0.0394 |
| Boujaad | 0.0172 | 0.0141 | 0 | 0.0761 | 0.0727 | 0.0778 | 0.0927 |
| Sardi | 0.0083 | 0.0054 | 0.0149 | 0 | 0.0234 | 0.0098 | 0.0149 |
| Timahdite | 0.0076 | 0.0049 | 0.0133 | 0.0057 | 0 | 0.0207 | 0.0285 |
| Beni Guil | 0.0038 | 0.0045 | 0.0133 | 0.0042 | 0.0048 | 0 | 0.0374 |
| Beni Ahsen | 0.0126 | 0.0079 | 0.0179 | 0.0062 | 0.0070 | 0.0078 | 0 |

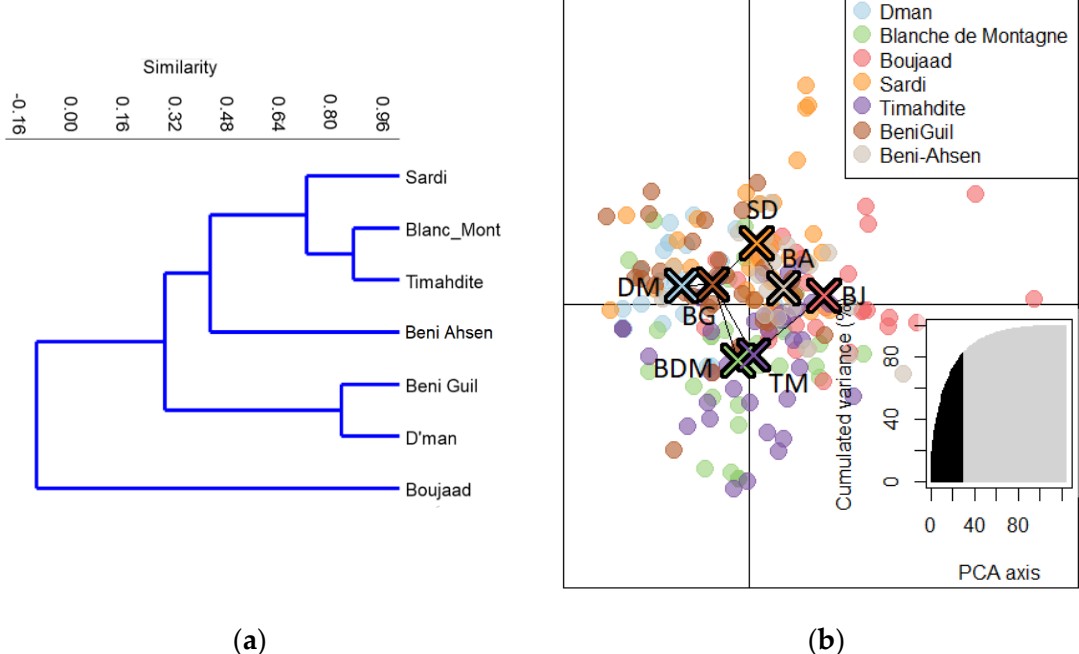

(**a**)　　　　　　　　　　　(**b**)

**Figure 2.** Relationships between Moroccan breeds: (**a**) cluster analysis of Nei's genetic distances using Chord as a measure of distance; (**b**) discriminant analysis of principal components (DAPC) scatterplot. Dots represent individuals, with colors denoting the breed allocation to clusters. The percentages of cumulated variance explained by principal component 1 (PC1) to PC30 are shown in the bottom left corner of the figure. The minimum spanning tree based on the (squared) distances between clusters within the entire space is shown. BA: Beni Ahsen; BG: Beni Guil; BDM: Blanche de Montagne; BJ: Boujaad; DM: D'man; SD: Sardi; TM: Timahdite.

From the approach of discriminant analysis of principal components (DAPC), the minimum spanning tree (Figure 2b) confirms that the closest breed to Beni Ahsen is Sardi. However, the Blanche de Montagne seems to be more distantly related.

3.1.3. Breed Assignation

DAPC also provided the probability of memberships of each breed (Figure S1). The grey color characteristic of Beni Ahsen is rarely spread in the other breeds and seems to be mostly present in Sardi, Timahdite, and Beni Guil. The average assignment probability is low at 0.545. Among the different breeds, Beni Ahsen and Timhadite showed the highest values around 0.6.

*3.2. Relationships with Foreign Breeds*

The whole dataset comprising 643 sequences covering 26 breeds (Table 1) was submitted to a network program and a phylogenetic analysis to describe the distribution and quantify the affinities between Merino, Merino-derived, and Moroccan breeds.

3.2.1. Network Approach

The sequences are arranged in 6 ensembles, corresponding to haplogroups A, C-E and B, composed of 1, 1 and 4 groups each, respectively (Figure 3). In this last haplogroup, the Moroccan breeds are distributed in two groups, close to Iberian and Italian breeds, named groups 1 and 2, respectively. Of note, the centers of the Iberian and Moroccan group 1 are occupied by several oriental animals (Tibetan). The Merino and Merino-derived breeds are included in the Iberian and Italian groups as follows: Merino preto, German Merino, and Transylvanian Merino in the Iberian group on one hand, and Merinizzatta, Gentile de Puglia, Sopravissana, and Spanish Merino in the Italian group on the other. As a result, there are two lineages of Merino breeds in Europe. As for the 20 Beni Ahsen sequences, their distribution is as follows: 14 in Moroccan group 1, 4 in the Italian group, 1 in Haplogroup A, and 1 in the Moroccan group 2.

3.2.2. Phylogenetic Approach

The numbers of sister sequences between the 26 breeds of the dataset were submitted to NMDS with correlation as a measure of distance. As shown in Figure 4, the Italian and Iberian Merino-derived are well separated, the Spanish Merino having the shortest distance from the Italian group. The non-Merino breeds of Italia are largely overlapping with the non-Merino, while the non-Merino breeds of Iberia occupy a large surface, overlapping both Italian and Iberian Merino-derived breeds. With regard to Moroccan sheep, five breeds are close to Iberian Merino: Beni Guil, Sardi, Beni Ahsen, D'man and Boujaad. The two remaining ones (Timahdite and Blanche de Montagne) are close to non-derived Merino breeds of Italia. Of note, the Swiniarka has a position close to the group of the five Moroccan breeds.

To answer the question of whether the Beni Ahsen is the closest breed to Merino-derived breeds, the proportions of sister sequences between Moroccan breeds and the Merino-derived ones were calculated (Figure 5). It shows that Beni Ahsen has the greatest number of direct connections (type 1) with Merino-derived breeds. At a closer look, these connections are only due to Transylvanian Merino and Merino Preto, i.e., belonging to the Iberian group. Beni Guil shares three-fourths of its connections with Merino breeds belonging to the Iberian group. As for connections of type 2, the three first Moroccan breeds—Beni Guil, Sardi, and Beni Ahsen—share 60%, 33%, and 85% with Merino-derived breeds belonging to the Iberian group, respectively. It can be concluded that 2 to 3 Moroccan breeds are connected to Merino-derived breeds mostly belonging to the Iberian group and that Beni Ahsen has a particular role.

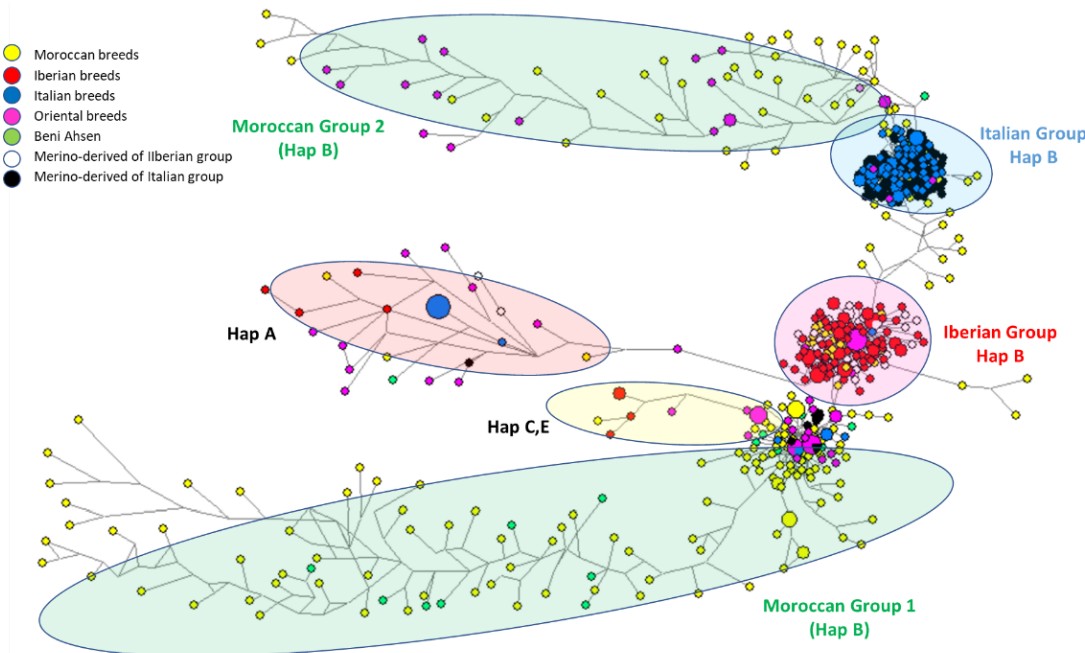

**Figure 3.** Network of Mediterranean s.l. breeds, including Beni Ahsen, among the two groups of Moroccan breeds. This figure was drawn using Network 10.2.0.0. The black and white circles correspond to Merino-derived breeds inside the Italian and Iberian groups, respectively. To test the connection between Merino-derived breeds and Beni Ahsen, we considered the tree generated by the ME method of phylogeny and its exploitation by the Newick-Extra program.

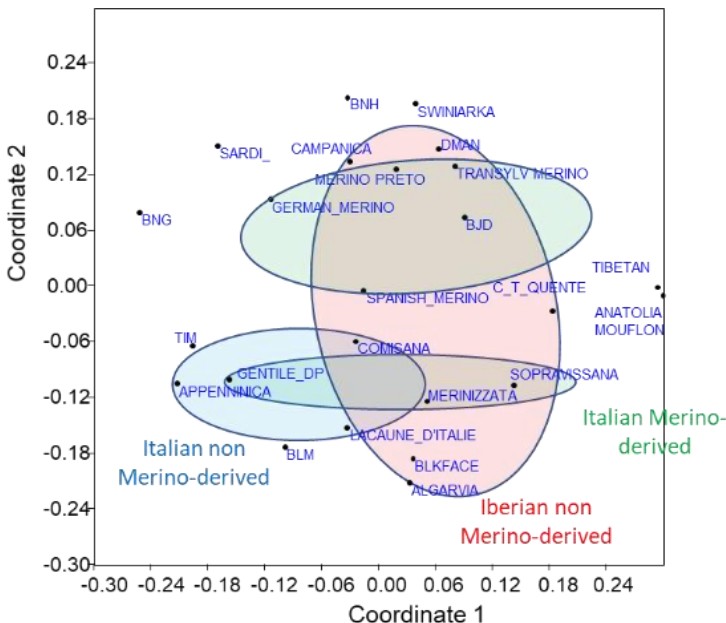

**Figure 4.** NMDS of sister sequence numbers between the 26 breeds. BA: Beni Ahsen; BG: Beni Guil; BDM: Blanche de Montagne; BJ: Boujaad; DM: D'man; SD: Sardi; TM: Timahdite.

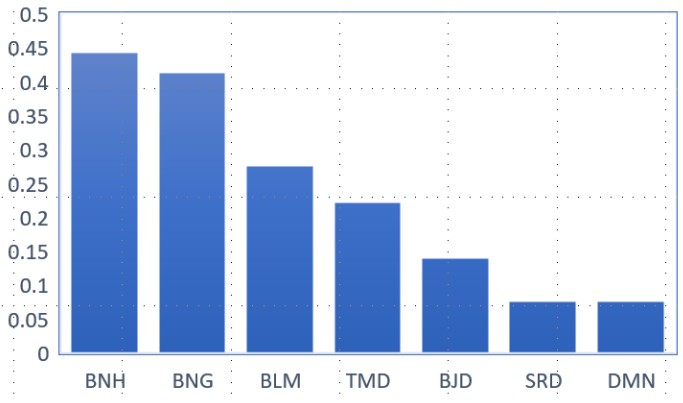

**Figure 5.** Proportion of sister sequence numbers between Moroccan and Merino-derived breeds. BA: Beni Ahsen; BG: Beni Guil; BDM: Blanche de Montagne; BJ: Boujaad; DM: D'man; SD: Sardi; TM: Timahdite.

### 4. Discussion

With regard to Beni Ahsen, it was shown that its closest Moroccan breed is Sardi. This link can be viewed as historical as these breeds belong to the animals of the great plains of west-center Morocco, according to Geoffroy Saint-Hilaire [6]. This view would go in the sense of recent analyses on breed differentiation, separating west and east breeds of the country [14]. Conversely, it could be due to recent crosses to improve the bad conformation of Beni Ahsen, but the calculations of membership probability do not favor this hypothesis, at least in our sampling. Indeed, in spite of its strong decline, the actual population of Beni Ahsen does not present an alarming level of genetic variability.

From the analyses of the mtDNA control region, it appears that the European Merino and Merino-derived breeds are clearly organized in two clusters, a Spanish one and an Italian one. According to the network analysis, the Spanish Merino settled in Estremadura in Spain would belong to the Italian group. This is supported by 10 connections of type 1 between Spanish Merino and Italian breeds, including 6 with Italian Merino-derived ones. However, from phylogenetic analysis followed by the Newick-Extra program, Spanish Merino rather occupies a median position between Italian and Iberian Merino-derived breeds. Maybe the network algorithm exaggerates the formation of clusters, unifying Spanish Merino in the sole Italian group. Anyway, the special position of Spanish Merino suggests a significant displacement of merinized ewes from Italy back to Spain. According to this hypothesis, this displacement would take place after the differentiation of the Merino breed in Spain. This could explain the great genetic polymorphism observed in Spanish Merino relatively to other Iberian breeds [23]. An opposite view would be a significant displacement of animals from Italy to Spain prior to the differentiation of the Merino type. According to a study based on SNP to Ciani et al. [24], there is a migration route from Balkan, Italia to Spain, maybe along the Mediterranean coast.

Interestingly, our results indicate that the two Merino-derived groups are more closely related to non-derived breeds of their respective countries than to each other, contrarily to the analyses deduced from another publication based on SNP by Ciani et al. [24]. These authors found that Merino-derived breeds form a single cluster close to Iberian breeds. Moreover, they indicated that the Italian-derived Merino breeds were very disconnected from the other Italian breeds. Is there a contradiction between these nuclear and mitochondrial markers? Not necessarily if the Merino character was mainly brought through Iberian males who were transported by ship to Italia and crossed to Italian ewes [17]. The analysis of the Y chromosome should shed some light on this issue.

Ciani et al. [24] demonstrated that an ancient stock of Merino had relationships with primitive animals. Here, it is interesting to note the presence of Tibetan individuals in the network center of Iberian and Moroccan group 1. Of course, the Tibetan animals themselves

must not be considered as the breed imported by humans in the West Mediterranean from the Middle East. They only represent primitive animals that are still close to the true ones that crossed the Mediterranean Sea, maybe with the help of Phoenicians or Romans, and took part of the genitors of both Moroccan and Iberian ovine populations. As shown by Pedrosa et al. [3], the Iberian sheep has certainly a complex origin from several immigration events. However, this author, as well as Ciani et al. [25], did not have information about North African genetic data.

Now, our new data bring some light on the enigmatic origin of Merino. The maternal genetic material indicates that Morocco has certainly contributed significantly. The Merino phenotype comprises, among others, the distribution of the fleece on the body (covering leg tips, forehead, and cheeks), the high density of hairs making waves, the chest dewlap, and the extreme thinness of wool. If the wool of Moroccan sheep is never as thin as in true Merino, most other characters can be found in several breeds, at least one century ago. Of course, whole-genome analysis using SNPs conducted on Beni Ahsen should enrich the present landscape because it offers the possibility of tracking the genes or at least chromosomal regions associated with these Merino-specific characters [26]. Nowadays, the Merino character is most recognizable in Beni Ahsen and to a lesser degree in Beni Guil and Boujaad. Our calculations indicate that the Beni Ahsen is the most genetically linked breed to Merino, especially its Iberian cluster. However, among Beni Guil and Sardi, some individuals are also connected to Iberian Merino. Due to selection, Sardi no longer shows the Merino phenotype except in its thin white wool, although some of its ancestors in the same territory did have it. It can be deduced that the extremely fine wool of present Merino was not directly given by Moroccan animals. Two non-exclusive hypotheses can be brought: (i) the animals involved in the introgression of this feature have now disappeared, and (ii) the selection by man during the Moor era was sufficient to reach the remarkable state appreciated in our times, as wool fineness is highly heritable. Indeed, it is easy to improve this character in a few generations through selection as it depends on a small number of mutations [26]. As Columella [4] wrote, the wool quality was brought by crossing with the Tarento (or Greek) breed during the Roman Empire. Interestingly, we found strong links between the Polish Swiniarka and the Moroccan breeds. This animal has primitive features and numerous defects of body conformation [27] but also fine hairs (22 to 26 μm) close to the standard hairs of Merino (around 21 μm, [28]). Moreover, it is undemanding in terms of feeding and highly resistant to diseases. Some of these characters were maybe present in the early founders of the Moroccan ovine population. To our knowledge, only the Blanche de Montagne looks a little like Swiniarka.

## 5. Conclusions

The landscape of Merino origin is substantially enriched, although the responses to numerous questions are pending. There is no proof but strong suspicion that animals from the Atlantic coast of Morocco, which should look like the present Beni Ahsen, were imported into the north part of the Strait of Gibraltar and crossed with local animals to enhance wool quality and production. The problem we are facing is that the Beni Ahsen is nearly extinct, preventing the possibility of further genetic investigations in the near future. Anyway, analysis of ancient DNA from both sides of the Strait of Gibraltar would be highly needed.

**Supplementary Materials:** The following supporting information can be downloaded at: https://www.mdpi.com/article/10.3390/ruminants2020013/s1. Figure S1: Probability of membership for each individual in the seven studied breeds. (**a**) Each column is representative of one individual and each breed is shown by a separate color. (**b**) At the right down corner are given the global probabilities of assignment. BA Beni Ahsen, BG Beni Guil, BDM Blanche de Montagne, BJ Boujaad, DM D'man, SD Sardi, TM Timahdite. Table S1. Phenotypic characteristics of Beni Ahsen breed, from [7,11]. Table S2. Access numbers for phylogenetic analysis.

**Author Contributions:** Conceptualization, I.B., D.P.P. and A.K.; software, A.K. and D.P.P.; resources: D.P.P., I.B. and M.P.; original draft preparation, A.K. and D.P.P.; writing—review and editing, I.B.; funding acquisition, D.P.P., I.B. and M.P. All authors have read and agreed to the published version of the manuscript.

**Funding:** This work was supported by the PHYLOSHEEP program of the AOI (University of Limoges).

**Institutional Review Board Statement:** As the blood samplings were made during routine medical care by a veterinary, ethic committee of IAV had issued an exemption letter for this study on 24 February 2022.

**Informed Consent Statement:** ANOC breeders consented to the utilization of the samplings for scientific purpose (Certificate established on 17 February 2022).

**Data Availability Statement:** The access numbers of all individuals considered in this study are in Table S1.

**Acknowledgments:** The authors are grateful to the technicians of the Veterinary Genetic Analysis Laboratory of IAV Hassan II and the Peirene Laboratory of Limoges University.

**Conflicts of Interest:** The authors declare no conflict of interest.

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
