# Peer review of "Genetic Diversity and Population Structure of Moroccan Beni Ahsen: Is This Endangered Ovine Breed One of the Ancestors of Merino?"

_ruminants, doi:10.3390/ruminants2020013_

Round 1
Reviewer 1 Report
The manuscript by Asmae Kandoussi et al. brings genetic data on Moroccan sheep to check if it is one of the ancestors to famous Merinos. The study is formally correct. However, it is obvious that the technique used in this manuscript is out of date. I also have some concerns about the manuscript as following.
1, It was stated that the polymorphisms of the Beni Ahsen were amplified by PCR. So it is not clear which parts of mtDNA were amplified? is it D-loop?
2, Is there any quality control evaluation for those mtDNA dataset? Is there any whole genome data in Genebank for those local sheep breeds, which is definitely more informative. I am pretty sure at least there are tons of genome sequence data for Merino sheep breed.
3, I am very confused about the figure5. What means sister sequences? How it was analyzed?
4, Is there any phenotype data like litter size/growth rate/coat colour et al. as supplemental materials to readers. It can make it more clear to the readers about the value of this study.
I would suggest that this manuscript should be done major revision before the second round of review.
Reviewer 2 Report
This paper is
- Using the control region of mtDNA Sequencing in 20 Beni Ahsen study survey was divided into Merino related breeds, genetic diversity was analyzed, and population structure was conducted.
- This paper, which performed Beni Ahsen (endangered breed) genetic diversity analysis in breeds are consider to be used as a very important data for genetic resource research of Merino.
- But, In my opinion, the title should be changed to “Genetic diversity and population structure of Moroccan breed (Beni Ahsen) ....”
- And, It would be good if the last part of the introduction was modified to be a normal sentence.
Reviewer 3 Report
General comments
The title sounds intriguing but do not reflect all the content of the paper since are reported mainly results on the genetic variability in the Marrocan breeds and their relation with other breeds.
Major remarks
lane 184 Figure 2 b. is not clear at all: it is a PCA or a DPCA? what means the X: the mean position of breeds? In this case BJ and not SD seems to be the closest breed to BA.
lane 198 The supplementary figure S1 could not be found, however the figure interpretation as well the explanatio at lane 339-342 were taken in account.
lane 267 how can be explained that Spanish Merino (the original one) groups with the italian breed (Merino and not Merino) and not with the other iberian (Merino and non Merino)?
lane 272 Can the discrepancy between the network and phylogenetic approach regarding the position of Spanish Merino better explained?
lane 333 Beni Ashen nearly exintc? In lane 64 are reported 385.000 animals in the 1999 (a number of animals by far above the treshold for risk of extinction), how is the present situation?
Minor remarks
lane 162 the legenda of the table do not correspond to the table (P before SG)
lane 169 the reported values 0,00198 and 00294 are not in the table
lane 216 Merinizzata instead of Merinizzatta
lane 235 Italy insted of Italia
lane 255 The code for the breeds in the legenda do not correspond to the figure 5
Reviewer 4 Report
Kandoussi et al present an interesting paper investigating the potential ancestors of Marino sheep, a widespread and economically important breed. Using mitochondrial control region sequences the relationship between the Moroccan breed Beni Ahsen and other Mediterranean sheep breeds. Kandoussi et al show that Marinos are a diverse breed and that - consistent with patterns in other domesticated animals - the breed designation does not indicate monophyly. Moroccan Beni Ahsen sheep show closer affinity with Iberian Marinos than they do with Italian Marinos.
Some specific comments:
ln 37 - "It is unlikely..." - why is it unlikely?
ln 82 - "during follow up..." - follow up to what?
ln 90 - lab methods don't need to be described in detail, but some information should be available to readers without having to dig through the references; put the genome positions sequenced relative to the sheep mt reference genome, and name the sequencing platform used
ln 109 - "primitive or Oriental" - neither of these words sit well in modern English; perhaps ancestral instead of primitive, and if Oriental is meant as a geographical designation, just be specific about the geography
ln 295 - man is an inappropriate generic for human
ln 316 - "during the Moor era" - put dates for this era
The English is clearly not written by a native speaker, but there is nothing about the sentence construction that makes it difficult for an English speaking reader to understand.
Round 2
Reviewer 1 Report
I am satified with their revision. I would suggest that this manuscript with minor revision can be accepted.
Author stated that they had not the time to undertake analyses using new softwares, and they agree that the sequencing of the whole mt genome would bring stronger results than the only control region. Actually, the author didn't understand what I meant. Nowdays there are plenty of whole genome sequencing data for local sheep breeds in public database especially for Merino this kind of famous sheep breed. Given the fast development of state-of-the-art whole-genome resequencing technology, author can discuss the main advantages and disadvantages of using mitochondrial markers or whole genome sequencing data to trace the Merino origin.
Author Response
We thank you again for your constructive remarks. Following your suggestion, we added at line 332, after the description of the major Merino-specific characters, the following sentence:
Of course, whole genome analysis using SNPs conducted on Beni Ahsen should enrich the present landscape because it offers the possibility of tracking the genes or at least chromosomal regions associated to these Merino-specific characters [26].
Reviewer 3 Report
I have not further comments
Author Response
We thank you for your efforts.